The role of sex and age in the architecture of intrapopulation howler monkey-plant networks in continuous and fragmented rain forests

Benitez-Malvido Julieta 1 jbenitez@cieco.unam.mx jbenitez@iies.unam.mx
Martínez-Falcón Ana Paola 1 2
Dattilo Wesley 3
González-DiPierro Ana María 4
Lombera Estrada Rafael 4
Traveset Anna 5
1 Instituto de Investigaciones en Ecosistemas y Sustentabilidad, Universidad Nacional Autonoma de Mexico , Morelia, Michoacán , Mexico
2 Centro de Investigaciones Biológicas, Universidad Autónoma del Estado de Hidalgo , Pachuca, Hidalgo , Mexico
3 Red de Ecoetología, Instituto de Ecología AC , Xalapa, Veracruz , Mexico
4 Unidad Académica Multidisciplinaria Las Margaritas, Universidad Intercultural de Chiapas , Las Margaritas Chiapas , Mexico
5 Ecología Terrestre Group, Biodiversity and Conservation, Institut Mediterrani d’Estudis Avançats (CSIC-UIB) , Esporles, Mallorca , Spain
Buckley Hannah
Electronic publication date: 2016 Mar 14
Publication date: 2016
Volume: 4
Electronic Location ID: e1809
Received 2015 Dec 4; Accepted 2016 Feb 23
Copyright: ©2016 Benitez-Malvido et al.
Copyright year: 2016
Copyright holder: Benitez-Malvido et al.
License: This is an open access article distributed under the terms of the Creative Commons Attribution License, which permits unrestricted use, distribution, reproduction and adaptation in any medium and for any purpose provided that it is properly attributed. For attribution, the original author(s), title, publication source (PeerJ) and either DOI or URL of the article must be cited.
License URL: https://creativecommons.org/licenses/by/4.0/

Keywords: Age, Alouatta pigra, Consumer-resource interactions, Habitat fragmentation, Sex-classes, Nestedness, Age class, Sex class

Funding: Consejo Nacional de Ciencia y Tecnología CB2005-C01-51043 CB2007-79121 Universidad Nacional Autónoma de México IN206111 This research was supported by grants from Consejo Nacional de Ciencia y Tecnología (CONACyT, CB2005-C01-51043, CB2007-79121 to JBM) and Universidad Nacional Autónoma de México (UNAM, IN206111 to JBM). The Posgrado en Ciencias Biológicas (UNAM) and CONACyT are greatly acknowledged for providing a doctorate scholarship to AMG and UNAM for a post-doctoral scholarship to APM. This manuscript was partially written while JBM was on sabbatical at IMEDEA and supported by DGAPA (UNAM). The funders had no role in study design, data collection and analysis, decision to publish, or preparation of the manuscript.

==============================
We evaluated the structure of intrapopulation howler monkey-plant interactions by focusing on the plant species consumed by different sex and age classes in continuous and fragmented forests in southern Mexico. For this we used network analysis to evaluate the impact of fragmentation on howler population traits and on resource availability and food choice. A total of 37 tree and liana species and seven plant items (bark, immature fruits, flowers, mature fruits, immature leaves, mature leaves and petioles) were consumed, but their relative consumption varied according to sex and age classes and habitat type. Overall, adult females consumed the greatest number of plant species and items while infants and juveniles the lowest. For both continuous and fragmented forests, we found a nested diet for howler monkey-plant networks: diets of more selective monkeys represent subsets of the diets of other individuals. Nestedness was likely due to the high selectivity of early life stages in specific food plants and items, which contrasts with the generalized foraging behaviour of adults. Information on the extent to which different plant species and primate populations depend on such interactions in different habitats will help to make accurate predictions about the potential impact of disturbances on plant-animal interaction networks.

Introduction

Trophic interactions among species constitute a central topic in ecology (Petchey, Morin & Olff, 2009). Some studies have evaluated how feeding relationships vary within populations (Bolnick et al., 2003). Often within the same population we can find both more selective (those that feed off a few plant species) or more opportunistic (those that feed off many plant species) individuals (Bolnick et al., 2003; Araújo et al., 2008; Araújo et al., 2010; Pires et al., 2011). During the development and growth of an organism, food requirements often change quantitatively and qualitatively principally because of metabolic costs, sex- and age-related preferences and foraging ability (Stevenson, Pineda & Samper, 2005). In addition, consumer growth can be accompanied by shifts in habitat use, which may result in changes in food availability, constraining the consumer capacity to exploit different types of resources (Bolnick et al., 2003; Petchey, Morin & Olff, 2009). According to the “Optimal foraging theory,” individuals consume a subset of potential resources depending on the resource and individual traits; in this sense, individuals always eat the most valuable resources. When preferred resources are scarce, individuals can eat unutilized resources (Marshall & Wrangham, 2007; Araújo et al., 2008; Araújo et al., 2010; Araújo, Bolnick & Layman, 2011).

An important ecological interaction in the Neotropics occurs between primates and the plant species they consume and disperse (Rivera & Calmé, 2006). Primate species such as howler monkeys (Alouatta spp.) have a flexible diet (e.g., leaves, fruits, flowers, and bark) that allows them to persist in human-disturbed habitats (Marsh & Loiselle, 2003). Groups of howler monkeys including infants, juveniles and adults, like some other primate species (e.g., Chiropotes spp. and Saguinus spp.), are able to cope with changes in resource availability within fragmented habitats through behavioural adjustments (e.g., food choice and foraging activity) (Jones, 2005; Isabirye-Basuta & Lwanga, 2008). Recent studies have shown that the degree of dietary variation in A. pigra is affected by both environmental (i.e., forest fragment size) and social (i.e., group size) factors (Dias & Rangel-Negrín, 2015). In fact, the persistence of primate populations and/or species in forest fragments largely depends on their ability to adjust their diet (Rivera & Calmé, 2006).

Several studies have used tools derived from network analysis to describe the dietary variation found in populations of animals with a focus on individual-based plant-animal networks (Pires et al., 2011; Tinker et al., 2012; Cantor et al., 2013). Recently, it has been shown that intrapopulation primate-resource networks are highly nested: diets of specialist individuals are a subset nested within the diets of generalist individuals (Dáttilo et al., 2014). There is no information, however, about potential factors determining such network structure. In this study, we used a network approach to investigate the structure of individual-based howler monkey-plant networks and their underlying mechanisms. The application of network theory allows the recognition of non-random patterns of interactions in food webs (Bascompte & Stouffer, 2009) and, in our case, the identification of the role of each individual within a food web based on the roles of all individuals within preserved and disturbed habitat conditions (continuous and fragmented forests). Moreover, a network approach in the study of primate diets enables us to assess the level of selectivity of an individual towards using plant species in a resource-limited environment such as small forest fragments.

To answer the question of what is driving diet selectivity and nestedness in howler monkey populations, firstly we assessed differences in resource availability (i.e., sampling of food trees and lianas) between continuous and fragmented rain forests. Secondly, we hypothesized that nestedness in howler monkey-plant networks results from the most selective age and sex class (male and female infants) feeding on a subset of the broader diet of another age and sex class (male and female adults). We used howler monkeys’ age and sex class to analyse consumer-plant interaction, because these categories have shown differences in behaviour and foraging patterns in primates as well as in other mammals (Brent, Lehmann & Ramos-Fernández, 2011; Fuentes-Montemayor et al., 2009; Stevenson, Pineda & Samper, 2005). For instance, species in the Ateline have shown differences between the sexes in diet. Adult females of spider monkeys (Ateles geoffroyi) eat live and decaying wood (e.g., Licania platypus trees) more often than do adult males, possibly to satisfy their mineral (e.g., sodium and/or calcium) requirements during pregnancy and lactation (Chaves, Stoner & Arroyo-Rodríguez, 2012); by contrast, adult females of black howler monkeys (A. pigra) are less active and feed mostly on fruits of high energy content when lactating (Dias & Rangel-Negrín, 2015). Moreover, fruit selection could differ between sexes and age classes within primate populations, with adult individuals consuming the largest seeds/fruits within a plant species (e.g., Lagothrix lagothricha in Stevenson, Pineda & Samper, 2005). Considering the postulates of the optimal foraging theory, in the absence or scarcity of their preferred resources in forest fragments, howler monkeys might consume a subset of the plant species consumed in continuous forests, which maintains the nestedness in both habitat types. From a conservation viewpoint, this information is useful if certain habitat elements such as forest fragments are to be employed effectively in the conservation of primates attention will need to be paid to their diet requirements.

Methods

Study area and habitat types

The research was conducted at the Lacandon rain forest, Chiapas, in southeastern Mexico (16°07′58″N, 90°56′36″W, 120 m elev.). Forest conversion has reduced the original forested area (500,000 ha) by two-thirds in the last 40 years (De Jong et al., 2000). Nevertheless, this region encompasses the largest remaining portion of tropical rain forest in Mesoamerica (Medellín, 1994). The primary vegetation type is lowland tropical rain forest, reaching up to 40 m in height in alluvial terraces. The temperature averages 23.9 °C, and annual rainfall is 2881.2 mm (González-Di Pierro et al., 2011). The study was conducted in two areas of lowland tropical rain forest separated by the Lacantún River: the Marqués de Comillas region (MCR, eastern side of the river) and the Montes Azules Biosphere Reserve (MABR, western side). The protected area of the MABR consists of 3,310 km2 of mature undisturbed forest. We selected three forest fragments occupied by black howler monkeys within the MCR area (one fragment of 6 ha and two fragments of 3 ha in area). Each fragment has its own independent howler monkey group. Fragments were isolated by 1–7 km from each other. All fragments have been isolated from continuous forest for at least 20 years (González-Di Pierro et al., 2011). In the continuous forest within the MABR we selected three sites used by three different howler monkey groups that were separated by 2 km from each other. Although howler monkeys have been observed crossing cattle pastures in the study area, individuals in this study did not move between sites and/or habitat types (AM González-Di Pierro, pers. obs., 2006–2008).

Resource availability

In the two habitat types we sampled and identified all trees species (≥10 cm diameter at breast height) to determine if resource availability (food availability, Tutin et al., 1997; Doran et al., 2002) differed between habitats (fragments and continuous forest). Within each site (three fragments and three continuous forest sites), we randomly located ten 50 × 2 m transects (0.1 ha) to sample trees (following Gentry, 1982). We minimized edge effects by locating all transects at least 100 m from the edge. We calculated the importance value index (IVI) of each species within each habitat (Moore & Chapman, 1986), which is an overall estimate of the percentage of relative frequency of a plant species in the community. Differences in tree community attributes (i.e., tree species richness, tree abundance, number of food tree species and IVI) between continuous forest and fragments were analysed with t-tests after log(x + 1) or an arcsine transformation of the data (the latter in the case of IVI). To test if differences in tree species similarity (Jaccard’s coefficient) were related to geographical distances among transects of each study site, we performed Mantel tests (Sokal & Rohlf, 1995).

Howler monkeys and dietary composition

This research complied with protocols approved by CONANP care committee (Comisión Nacional de Áreas Naturales Protegidas) and DGVS (Dirección General de Vida Silvestre, permission number SGPA/DGVS/07830). The collection of vegetation and feeding behaviour data did not interfere with primates in any way. The black howler monkey (Alouatta pigra) is present in Mexico, Guatemala and Belize, but most (ca. 80%) of its distribution range is found in Mexico. It is one of the largest Mesoamerican primates. The conservation status of the species is “endangered” according to the IUCN Red List (http://www.iucnredlist.org/apps/redlist/search), and habitat loss is probably the most important threat affecting the populations. Howler population density within the MABR is 0.13 individuals/ha, but within the study fragments (3–6 ha, MCR) population density averaged 1.3 individuals/ha. Home-range size of black howlers in continuous forest is <25 ha (Estrada, Van Belle & García del Valle, 2004).

Dietary composition of howler monkeys was studied during a period of 18 months: three months in the dry season from February to April of 2006, 2007 and 2008 and three months in the rainy season from August to October of the same three years. We did not examine between seasons and/or year changes in the food availability for primates because we needed a large and complete data set in which all plant species and age and sex classes were represented to construct the ecological networks. Feeding behaviour was documented during three consecutive days once every three weeks, using five minutes of focal animal sampling (Altmann, 1974; Martin & Bateson, 2007). Each individual was recognized by characteristically unique marks on their skin and hair. Monkeys were systematically observed from 7:00 am to 17:30 pm.

At the beginning of the study, we categorized the focal individuals by age and sex class into six groups as follows: adult male, adult female (adults are full-grown individuals with conspicuous sexual organs; males have an enlarged, noticeable hyoid bone); juvenile male and female (juveniles are completely independent from adult females but not yet full-grown); and male and female infant (infants depend on their mothers for locomotion and feeding, in some instance). To construct the ecological networks, individuals were kept in their initially designed age and class category despite the fact that infants were more independent at the end of the study. In continuous forest, we recorded 15 individuals: six adult females, four adult males, two juveniles (female and male), two infant females and one infant male. In forest fragments, we recorded a total of 18 individuals: five adult females, four adult males, four juvenile females, three juvenile males, one infant female and one male. Howler monkey population size and structure remained unchanged in fragments and continuous forest during the course of the study. There was no birth or death in any of the studied groups.

All howler monkey individuals were observed for the same period of time in each habitat type. The effect of habitat on feeding time and on the usage of different plant items was analysed by comparing the fraction of time spent consuming different plant items (i.e., flowers, petioles, young and mature leaves, mature and immature fruits and bark) within continuous forest and forest fragments with a nested-ANOVA of angular transformed data. Data were analysed using the statistical program SigmaStat for Windows 3.5. Furthermore, we refer to a preferred food (i.e., an over selected food) as those plant species and items selected (usage) disproportionately often relative to their abundance (availability) and to a fallback food as those plant species and items that howler monkeys utilized when preferred foods are scarce (Marshall & Wrangham, 2007). Typically, fallback foods are plant species and items of low preference but high importance in the diet (e.g., liana leaves). Plant items of high importance are those most frequently consumed regardless of their nutritional quality; whereas preferred items are those of high quality, with quality defined as rate of energy return to an organism (e.g., ripe fleshy fruits).

Network metrics

We used the NODF metric (nestedness metric based on overlap and decreasing fill, Almeida-Neto et al., 2008) to evaluate whether or not the diets of more selective monkeys represent subsets of the diets of monkeys that consumed a broader based diet for each habitat. Because not all age and sex classes were present in all sites, we pooled individuals present within each habitat type (fragments and continuous forest) to construct the networks from an intrapopulation perspective. NODF is recommended in ecological network analysis because it is less prone to type I errors (Almeida-Neto et al., 2008). We generated theoretical matrices to test the significance of the nestedness observed against null distributions of these values generated by the Null Model II (Bascompte et al., 2003) in ANINHADO software (Guimarães & Guimarães, 2006). We generated random matrices to test the significance of nestedness according to the Null Model II by using functions within the software ANINHADO (n = 1,000 randomizations for each network). In this null model, the probability of occurrence of an interaction is proportional to the number of interactions of both plant species and monkey individuals (Bascompte et al., 2003). In our intrapopulation networks, plant species and monkeys are depicted as nodes, and their feeding interactions are depicted by links describing the use of plant species by individuals. Our qualitative approach in calculating nestedness decreases the probability of overestimating the amount of resources (e.g., leaves vs. fruits) ingested by monkeys (Dáttilo et al., 2014). Biologically, nestedness describes the organization of niche breadth in which more nested networks tend to have the highest niche overlap (Blüthgen, 2010).

Other network parameters considered in the analysis were as follows: (i) mean linkage level (mean number of links/interactions per species); (ii) connectance (the proportion of realized links of the total possible in each network, defined as the sum of links divided by the number of cells in the matrix); (iii) interaction diversity (based on the Shannon diversity index); and (iv) resource selectivity at the network level (H2′). This selectivity index ranges from 0 (extreme generalization) to 1 (extreme specialization) and is extremely robust with changes in sampling intensity and the number of interacting species (Blüthgen, 2010). Network features were estimated with the Bipartite package (Dormann, Gruber & Group, 2011). Network plots were obtained by using Bipartite in ‘R’ (Dormann, Gruber & Group, 2011; R Development Core Team, 2011).

The categorical core vs. periphery analysis was used to describe plant species as core (generalist species, those with the most interactions) or peripheral (those with fewer interactions) components of the network. Core–periphery analyses were performed with UCINET for Windows 6.0 (Borgatti, Everett & Freeman, 1999), which performs two routines for detecting core–periphery structures in bipartite graphs (n = 20 runs/network) and obtains the percentage of occurrence of core–periphery species (see Borgatti et al., 1999; Díaz-Castelazo et al., 2010).

Results

Overall, we found that important food resources, including plant species and items, changed with habitat type, and age and sex classes indicating that forest fragmentation affects the feeding behaviour and level of resource selectivity of howler monkey populations in our study sites.

Resource availability

Continuous forest and fragments presented similar tree species richness and density (diameter at breast height > 10 cm), similar numbers of tree species consumed by howler monkeys and a similar IVI of food species (for all cases t < 2, df = 5, P > 0.05; Table 1). Tree species’ similarity (Jaccard’s coefficient) between continuous forest and fragments was ca. 70%. The Mantel test showed no significant association between tree species similarity and geographical distances within and between habitat types (t∝ = 0.57, P = 0.60): species were as likely to be found in 0.1 ha blocks close together as in those far apart. Fragments and continuous forest shared 50% of the 10 tree species with the greatest importance value index (IVI), all of which are consumed by howler monkeys (Table 2).

Table 1 Tree community (diameter at breast height > 10 cm) attributes in continuous forest and forest fragments inhabited by howler monkeys (Alouatta pigra) in the Lacandonian rain forest, Chiapas, Mexico.

The values are the average (±SD) of ten 50 × 2 m transects (0.1 ha) in each of three forest fragments and three continuous forests. Tree community attributes did not differ significantly between habitat types (for all cases t < 2, df = 5, P > 0.05).

Tree attributes	Continuous forest	Forest fragments	
Mean tree species richness (±SD)	33.7 (4.7)	33.3 (2.1)	
Mean number (±SD) of primate-dispersed tree species	16.0 (3.1)	12.0 (0.6)	
Mean density of trees (dbh > 10 cm)	141 (16.4)	137 (5.5)	
IVI of food species	6.7	6.5	
Notes.

The importance value index (IVI) was calculated by summing the density, the frequency and basal area of each species within each habitat (Moore & Chapman, 1986).

Table 2 The ten tree species with the highest importance value index (IVI) in continuous forest and forest fragments occupied by howler monkeys (Alouatta pigra) at the Lacandonian rain forest, Mexico.

All tree species are present in the diet of howler monkeys.

Family	Species	IVI	
Continuous forest			
Moraceae	Brosimum alicastrum	0.52	
Meliaceae	Guarea excelsia	0.40	
Moraceae	Ficus sp.	0.36	
Ulmaceae	Ampelocera hottlei	0.22	
Burseraceae	Bursera simaruba	0.19	
Anacardiaceae	Spondias mombin	0.15	
Moraceae	Trophis racemosa	0.15	
Fabaceae	Acacia usumacintensis	0.13	
Moraceae	Castilla elastica	0.12	
Fabaceae	Albizia leucocalyx	0.11	
Forest fragments			
Fabaceae	Dialium guianense	0.53	
Moraceae	Brosimum alicastrum	0.46	
Fabaceae	Pterocarpus bayesii	0.28	
Ulmaceae	Ampelocera hottlei	0.26	
Moraceae	Ficus sp.	0.21	
Moraceae	Castilla elastica	0.19	
Chrysobalanaceae	Licania platypus	0.18	
Sapotaceae	Pouteria campechiana	0.16	
Moraceae	Trophis racemosa	0.15	
Meliaceae	Guarea excelsia	0.11	

Howler monkey dietary composition

Overall, the total time spent making focal observations in fragments was 167.30 h but was 146.66 h in continuous forest because there were more individuals in fragments. Howler monkeys inhabiting forest fragments spent more time foraging (61.22 h or 36.74% of the time) than those monkeys inhabiting continuous forest (39.55 h or 26.52% of the time). Adults and juveniles in forest fragments spent more time foraging (adults, 46.14 h; juveniles, 14.96 h) than adults and juveniles in continuous forest (adults, 31.30 h; juveniles, 5.56 h); whereas in continuous forests infants spent more time foraging (2.80 h) than infants present in forest fragments (0.12 h). We found 30 plant species consumed for all age and sex classes in forest fragments and 27 in continuous forest (Fig. 1 and Table 3). A total of 37 plant species and seven plant items (i.e., bark, immature fruits, flowers, mature fruits, immature leaves, mature leaves and petioles as in Table 3) were consumed in both habitats. These included 32 species of trees, four species of woody lianas (Abuta panamensis, Bignonaceae sp., Macherium sp. and Malpighiaceae sp.) and one species of a climbing herb (Araceae sp.). The time devoted to consuming different plant items was similar for both habitats (F1,13 = 0.53, P = 0.49), while plant items within habitats were consumed with significantly different frequency (nested-ANOVA F6,13 = 13.13, P = 0.003). Overall, the plant items consumed with significantly greatest frequency (number of records per feeding time) were mature fruits and immature leaves for both habitat types (Fig. 1). Feeding time changed among plant items between habitat types as follows: mature fruits in continuous forest, 54.5% vs. 37.6% in forest fragments; immature leaves, 31.0% vs. 56.2%; immature fruits, 1.2% vs. 5.0%; petioles, 6.2% vs. 0.5%; mature leaves, 3.8% vs. 0.8%; and finally bark, 2.3% and flowers, 0.73% only in continuous forest. Not all items were consumed in all plant species and habitats (Table 3); in continuous forest flowers (i.e., Machaerium sp.) were only consumed by females of all ages, whereas bark (i.e., Licania platypus) was only consumed by adult and infant females and by adult males. In continuous forest, howlers spent more time eating mature fruits (more than 50%) followed by immature leaves (31%), regardless of age-class. By contrast, in fragments, adults and juveniles of both sexes spent more time consuming immature leaves (50%) followed by mature fruits (30%), whereas infants spent all of their time eating immature leaves.

Figure 1 Diet composition of howler monkeys (Alouatta pigra) in continuous forest and fragments according to percentage of total feeding time consuming different plant items.

Significant (P < 0.05) differences in the consumption of plant items is indicated with an asterisk (*). The items consumed per plant species are indicated for each habitat type; where items are: B, bark; IF, immature fruit; FL, flower, MF, mature fruit; IL, immature leave, ML, mature leave and P, petiole.

Table 3 Occurrence of plant species in the core and in the periphery for each network (continuous forest and fragments).

The items consumed per plant species are indicated for each habitat type. Plant items per species are arranged from left to right, with items at the far left being the most consumed; where items are: B, bark; IF, immature fruit; FL, flower; MF, mature fruit; IL, immature leave; ML, mature leave and P, petiole.

Plant species	Continuous forest	Forest fragments	
	Item	% core	% periphery	Item	% core	% periphery	
Brosimum alicastrum	IL, MF, IF	100	0	IL, MF, IF	100	0	
Ficus sp.	–	0	0	IL, MF, IF	100	0	
Abuta panamensis	MF	0	45	MF, IL	100	0	
Acacia usumacintensis	IL	0	45	IL	95	5	
Ampelocera hottlei	MF, ML	100	0	MF	100	0	
Araceae sp.	–	0	0	IL	95	5	
Bignoneaceae sp.	IL	0	100	IL, MF	100	0	
Cecropia obtusifolia	MF, IL	85	15	IL, MF	95	0	
Cojoba arborea	IL	0	100	IL	100	0	
Dialium guianense	MF	100	0	MF	100	0	
Licania platypus	B, IL	45	55	IL	100	0	
Machaerium sp.	IL, FL, MF	0	100	IL	100	0	
Pourouma bicolor	MF	100	0	MF, IL	90	10	
Trophis racemosa	–	0	0	IL	100	0	
Brosimum lactescens	MF	100	0	MF	25	75	
Castilla elastic	MF	0	100	IL	40	60	
Combretumsp.	–	0	0	IL	0	80	
Hirtella Americana	–	0	0	MF	30	0	
Liana sp.	–	0	0	IL	20	80	
Paulinia fibrigera	IL	50	50	IL	20	80	
Pseudolmedia oxyphillaria	–	0	0	IL	30	70	
Talauma Mexicana	P	0	100	P	0	90	
Albizia leucocalyx	ML, IF	100	0	IL	0	100	
Garcinia intermedia	MF	95	5	MF	0	100	
Inga sp.	IL	60	0	IL	0	100	
Platimiscium yucatanum	IL	0	45	IL	0	100	
Sapindaceae sp.	-	0	0	MF	0	100	
Schizolobium arboreum	P	0	100	P	0	100	
Spondia mombin	-	0	0	MF	0	100	
Bursera simaruba	–	0	0	IL	0	100	
Ficus tecolotensis	MF, IL	100	0	–	0	0	
Bravaisia sp.	IL, P	55	45	–	0	0	
Maclura tinctoria	IL	100	0	–	0	0	
Ficus yoponensis	IL	0	100	–	0	0	
Lonchocarpus sp.	ML	0	100	–	0	0	
Malpigiaceae sp.	FL	30	70	–	0	0	
Zanthoxylum riedelianum	IL	0	100	–	0	0	

Howler monkey-plant networks

We found a significant nested pattern in our howler monkey-plant network in both continuous forest (observed matrix: NODF = 51.41; mean ± SD of simulated matrices: NODF = 44.78 ± 3.48; P = 0.04) and fragment habitats (observed matrix: NODF = 62.42; mean ± SD of simulated matrices: NODF = 45.71 ± 2.89; P = 0.01) (Fig. 2 and Table 4). Network attributes for the two habitats presented similar values of connectance, links per species, interaction diversity and resource selectivity. We found, however, lower links per species in continuous forest than in forest fragments, which probably generated greater resource selectivity and specialization in continuous forest (Table 4).

Figure 2 Intrapopulation howler monkey-plant networks (Alouatta pigra) for (A) continuous forest and (B) forest fragments.

Each node represents one monkey (left) or plant species (right) and lines represent monkey–plant interactions. Codes for A. pigra age-classes are the following: light box, adult female; dark box, adult male; light triangle, juvenile female; dark triangle, juvenile male; light diamond, infant female; and dark diamond, infant male.

Table 4 Howler monkey-plant network attributes in continuous forest and forest fragments at the Lacandon rain forest, Mexico; see methods for details.

Network metrics	Continuous forest	Forest fragments	
No. of monkeys	15	18	
No. of plant species	27	30	
Nestedness (NODF-metric)a	51.41	62.42	
Links per species	3.76	4.08	
Connectance (C)	0.39	0.36	
Interaction diversity	5.06	5.27	
Resource selectivity H2′	0.28	0.22	
Notes.

a Both networks were significantly nested (P < 0.05).

Species turnover as core/periphery components in fragments and continuous forests networks was very high as plant species fluctuated between habitats as core or periphery components (Table 3). There were, however, three strict core species (i.e., A. hottlei, B. alicastrum and D. guianense) and one strict peripheral species (i.e., Schizolobium arboreum). The liana species Bignonaceae sp., Macherium sp., and the tree Cojoba arborea were core species in fragments but periphery in continuous forest, whereas the tree species Albizia leucocalyx, Brosimum lactescens and Garcinia intermedia were core species in continuous forest but periphery in fragments. Not all common food tree species were those preferred by howler monkeys in the network analysis (Tables 2 and 3). For instance, Ficus sp. was a core species in forest fragments and peripheral in continuous forest, whereas Ficus tecolotensis was core in continuous forest and peripheral in forest fragments. Moreover, P. bicolor was a core species for continuous forest and fragments; however, it is not within the 10 tree species with the highest importance value index (IVI) in either habitat (Table 2).

Discussion

Overall, we observed that resource choice in fragments was lower within howler monkey populations, despite the presence of preferred plant food species for primates in both habitat types; their relatively low selection may have been driven by habitat attributes such as the relative scarcity of the most favoured feeding plant species and items of forest fragments (Dias & Rangel-Negrín, 2015). Furthermore, we found a novel pattern of age, sex diet composition variation, indicating the presence of a sex, age class selectivity in the interaction between howler monkeys and the plant species they consume. This study is the first to show that age and sex classes determine the structure of ecological networks in primate-plant interactions. Regardless of habitat type, howler monkey populations are composed of both more selective and less selective individuals (Fig. 2). In this monkey-plant system we have shown that less selective individuals (i.e., adult males and females) consumed large amounts of resources independent of type and availability, thus building a cohesive network to which more selective individuals were attached (i.e., male and female infants) (Bascompte et al., 2003). Thus, as “generalist” consumers, adults maintain the stability of the network.

Habitat, food choice and availability

Plants and animals contributed to the nested pattern in both habitat types. The high plant species turnover as core–periphery between continuous forest and fragments was evident in the consumption of Abuta panamensis, Bignonaceae sp., Brosimum lactescens, Cojoba arborea and Macherium sp. (Table 3). All except B. lactescens were core species in fragments and peripheral in continuous forest. One unidentified species of Ficus sp. had a higher importance value index in both habitats; however, howler monkeys consumed it more often in fragments than in continuous forest. Several fig species are common and very important in the diet of several Neotropical primate species in different habitat types (Chaves, Stoner & Arroyo-Rodríguez, 2012; Dáttilo et al., 2014).

The preferred plant species and items in continuous forests are limited or unavailable in fragments. Therefore, howler monkeys in fragments may rely on resources of relatively low preference (fallback foods) to fulfil their nutritional requirements (Marshall & Wrangham, 2007). Plant parts or items of liana species as well as immature fruits and leaves of tree species were more frequently consumed in fragments than in continuous forest (Fig. 1). In preserved forests, howler monkeys are known to select large ripe fruits and immature leaves that are more easily digested (Leighton, 1993; Behie & Pavelka, 2015). Lianas in fragments, by contrast, are typically abundant and important in the diet of howler monkeys, but some of their plant parts or items may provide low rates of energy gain when compared to preferred foods. Forest fragmentation is known to increase the mortality of large fruit trees, to favour the proliferation of several liana species and to negatively affect tree phenology (reduced fruit set; Laurance et al., 2001; Chaves, Stoner & Arroyo-Rodríguez, 2012).

Fragmentation affects the availability of mature fruits to primates through reduction in the abundance and richness of large food trees, as larger trees produce more fruits than smaller ones (Chapman et al., 1992; Laurance et al., 2001; Chaves, Stoner & Arroyo-Rodríguez, 2012). The decreased richness and abundance of large trees could negatively affect the distribution and abundance of many tropical primates, especially in the case of highly frugivorous species (Chapman et al., 2007). A large proportion of tropical tree species produce fleshy fruits, allowing a year-round offer of resources that maintains several species of frugivores (Howe & Smallwood, 1982; Fleming & Kress, 2013). However, substantial changes in resource availability—both temporally and spatially—within fragments may prevent howler monkeys from searching for and consuming their preferred plant items and species (e.g., ripe fruits).

Network attributes (nestedness, connectance, mean linkage density, interaction diversity) were similar between habitats. The higher consumption of preferred items in continuous forest might arise because howler monkeys are not limited and have the possibility to range freely and feed on the best resources (i.e., ripe fruits). In fragments, by contrast, they have to consume what is available, which may represent a restricted set of food choices (resulting in a greater overlap of plant species consumed items) causing the monkeys to spend more time feeding in fragments to fulfil their nutritional needs.

Age and sex class, plant items and habitat

Our findings indicate that individuals do not forage randomly when compared to null models and that the diets of more selective monkeys (infants) represent subsets of plants and items consumed by other group members (adults), implying that individuals differ in their foraging strategies. Adult individuals are able to consume a wide range of plant species and items and therefore make the strongest contribution to the nested structure of the system. Infants may become generalists as they learn how to eat a wider range of plant species. Howler monkey infants tended to be more selective, while juveniles consumed a more diverse set of plant species than infants did. Adults, though, consumed the greatest variety of plant species and items. Male and female infants tended to consume more plant species in continuous forest than in fragments, whereas adult females were the most extreme “generalists” in the resulting networks (Dáttilo et al., 2014). This study is the first one to show that the specialized diets of male and female infants determine the nested structure of primate-plant networks in howler monkey populations.

Plant species making the greatest contribution to community nestedness (promoting asymmetry) were those species yielding greater fruit supplies and therefore a greater number of interactions and greater plant items consumed by howlers (Table 3). These strict core species were not necessarily the most abundant in either habitat. According to the optimal foraging theory (Araújo et al., 2008; Araújo, Bolnick & Layman, 2011), the individual niche depends on the availability of resources in the habitat; we indeed found that howler monkeys in fragments consume resources that were not utilized in continuous forest (e.g., lianas).

Conclusion

We aimed to understand the intrapopulation factors affecting the feeding ecology of howler monkeys and to link habitat fragmentation and howler monkey-resource interactions by using a network approach. Our findings indicate that both age and sex class drive a nested pattern in howler monkey-plant interactions; in this sense, this study is the first to provide a mechanism that structures such networks. Furthermore, we found that within howler groups, adults—particularly female adults—are likely to be the main seed dispersers for several old-growth forest tree species (e.g., seeds > 1 cm in length) in the study region because their diet consists primarily of mature fruits (Behie & Pavelka, 2015; Dias & Rangel-Negrín, 2015). In the Lacandon forest, we were able to detect that howler monkeys inhabiting fragments displayed less dietary selection because of the limited availability of preferred food readily available in continuous forest, which may threaten their long-term persistence in disturbed habitats.

Supplemental Information

Supplemental Information 1 List of plant species

List of plant species consumed by different age and sex classes of howler monkeys (Alouatta pigra) in continuous forest and tropical rainforest fragments of the Selva Lacandona, southeast Mexico.

Click here for additional data file.

We warmly thank J M Lobato, H Ferreira and A Valencia for careful technical support.

Additional Information and Declarations

Competing Interests

Author Contributions

Animal Ethics

Field Study Permissions

Data Availability

Anna Traveset is an Academic Editor for PeerJ.

Julieta Benitez-Malvido conceived and designed the experiments, performed the experiments, contributed reagents/materials/analysis tools, wrote the paper, prepared figures and/or tables, reviewed drafts of the paper.

Ana Paola Martínez-Falcón and Wesley Dattilo analyzed the data, wrote the paper, prepared figures and/or tables, reviewed drafts of the paper.

Ana María González-DiPierro conceived and designed the experiments, performed the experiments, analyzed the data, contributed reagents/materials/analysis tools, prepared figures and/or tables, collected and processed the data bases.

Rafael Lombera Estrada performed the experiments, collected de primate and vegetation data; followed the monkey troops.

Anna Traveset wrote the paper, reviewed drafts of the paper.

The following information was supplied relating to ethical approvals (i.e., approving body and any reference numbers):

This research complied with protocols approved by CONANP care committee (Comisión Nacional de Áreas Naturales Protegidas, permission number SGPA/DGVS/07830). The collection of vegetation and feeding behaviour data did not interfere with primates in any way.

The following information was supplied relating to field study approvals (i.e., approving body and any reference numbers):

CONANP (Comisión Nacional de Áreas Naturales Protegidas, permission number SGPA/DGVS/07830).

The following information was supplied regarding data availability:

The research in this article did not generate any raw data.

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
