# Peer review of "The role of sex and age in the architecture of intrapopulation howler monkey-plant networks in continuous and fragmented rain forests"

_PeerJ, doi:10.7717/peerj.1809_

## Round 0.1 · original submission · Major Revisions

Please pay close attention to all reviewers comments and address each explicitly, showing how you have responded to the comment in the manuscript. I strongly advise that you have the manuscript edited for English usage and readability.

Reviewer 1 ·

Basic reporting

L132: Should “geographic” be replaced with “geographical”?
L155: It is mentioned here that “…population size and structure remained unchanged…during the course of the study”. Is this an observed phenomenon of the study in the study area or an assumption employed in the study? Clarification would be needed.
L165: Full phrase for NODF is not given in the paper.
L172: Full phrase for CE is not available in the paper.
L276: The format of reporting for “…(fallback foods; sensu Marcshall & Wrangham, 2007)” is confusing and should be improved so that relationship among the “fallback foods”, “sensu” and “Marcshall & Wrangham, 2007” could be made clear to the reader.
L439: Table I – The title of the table is “Tree community (…) attributes …”, however, the 1st column panel heading title is “Vegetation attributes” which is not tally with the table title. Should it be “vegetation attributes” or “tree community attributes”? If they mean the same in this study, it is better to use a standard term throughout the paper.
The section on "Conclusions" could NOT be found in this paper.

Experimental design

L151 - 154: 15 and 18 individuals of howler monkeys are recorded in the continuous and fragmented forests, respectively, in this study. However, method on how to make sure that one individual (e.g. female adult) observed is uniquely different from the other individuals (other female adults) identified in the same habitat is not mentioned. Please briefly provide the method used in this aspect, if any.
L167: The meaning of this sentence is not clear. What does it mean with “…not all age and sex classes were present in all sites…”? Does it mean that apart from the 18 and 15 recorded individuals in the study; there are many other individuals available in each habitat? Clarification would be needed. Whether the term “site” refers to “habitat type” is not clear as well. Standardization of the use of key terms in the study like habitat would be helpful in understanding the study carried out.
Assumption employed in the study is not obvious in the paper. It will be good to briefly discuss those assumptions explicitly, e.g. whether individuals in one habitat visit and consume resources in other habitat and etc.

Validity of the findings

L221-225: Implication for the finding discussed here is not clearly explained in the discussion. Apart from this, the decline as claimed here is not consistent as, for example, the percentage for mature fruits in forest fragments is 37.6 whereas it is 56.2% for immature leave, which is an increase instead of a decrease.
L229: What does the percentage reported here refer to? If it refers to time, the time devoted for immature leaves in continuous forests is 31% (as given in L223) instead of 20% (as stated in this sentence).
L239 – 240: From Table IV, values for connectance (with a difference of 0.03), links per species (difference of 0.32) and interaction diversity (difference of 0.21) are commented as similar in these sentences for the two habitats. However, value for nestedness (H2’) with a difference of 0.06 is regarded as “…showing significantly lower…”. Further clarification may be needed to explain the reason of interpreting the values of the network metrics differently because even though the difference for values of links per species and interaction diversity is higher, they are commented as ‘similar’ when compared to the small difference for H2’ which is commented as ‘significantly lower’.
L256: What does “unprecedented pattern” refer to? Please explain.
L272: With regards to the “…spatio-temporal availability”, the ‘temporal’ aspect is not clearly explained.
L289: It will add value and further justify to the explanation given in this sentence if there is any past survey result to be compared, especially those that reporting the similar result.
L308: The meaning of this sentence is not clear – does it mean that “the individuals in fragments go over to the continuous forest and consume unutilized resources in the continuous forest”? Discussion on the interaction between the individuals at a habitat with the resources in the other habitat could hardly be found in the paper. Clarification would be needed.
L314: The source from where “…particularly adult females…” is claimed as stated in this sentence is not given.
L317: The reason of bringing up the issue of “seed dispersers” in the last paragraph of the section of discussion is not clear. Having this as the question this study intends to answer is not mentioned in the paper.
L319: The reason of making this statement need further explanation as to how it is implied from the findings obtained in this study.

NO conclusion is made with regard to the study conducted. The last paragraph in the section of discussion seems to open up new issues like “seed disperser” and “threatened long-term persistence” that need further explanation rather than concluding the whole study.

Additional comments

This is an interesting study on an endangered species – howler monkey. According to the paper, the objectives of the study is (i) to assess differences in resource availability between the continuous and fragmented forests (L89); and (ii) to evaluate its hypothesis on nestedness as a result from the most selective age, sex-class (infants) feeding on a subset of the broader diet of another age-sex class (adults) (L90). These are mainly to answer the question of what is driving diet selectivity and nestedness in howler monkey populations (L88). The paper has shown that these two purposes have been achieved and thus the research question is answered. However, no conclusion is made with respect to the purposes the study intends to achieve and the gap the study wants to fill. In addition, it will be good for the paper to have someone to proof-read for the language.

Reviewer 2 ·

Basic reporting

The article does not include sufficient introduction and background to support the relevance of the study. At this part, there are few studies that refers to the howlers (focusing in social structure matrix, seed dispersal, and only one about diet), however the sentences that the authors refer in those papers, are not related to the main article findings. Considering that Alouatta genus, is one of the most studied neotropical primates it will be necessary to include more significant references, this will improve the work.
The hypothesis that the authors are testing, is not relevant enough, mainly because they are not considering some important aspects about black howlers biology.

Experimental design

The methods are not described with congruent information, in some parts of the text it has been described that howler monkeys population remained unchanged in both study sites (see, howler monkeys and dietary composition), but this is not clear if it is consider that this study was made during three years, the authors do not mention mortality or births, and the categories (infant and juvenile), certainly changed according to the years. This is one of the main concerns, because it is not well describe.
The authors studied the howler population during the dry and rainy season, however they do not made any analysis concern to the feeding variations for each season. This should be consider because rainfall is associated with habitat productivity and seasonality, and therefore food availability.
About ethical standards, the permission is made by the "Dirección General de Vida Silvestre" DGVS, and not by CONANP. this should be changed.

Validity of the findings

The main concern about the findings is that they are not well supported, there are missing comparison between the findings found in the present study and some other studies made under the same line of knowledge. This should be solved making comparison with some other articles, that consider some biological aspects (i.e growing and development in infants, differences in females stage, lactation, pregnancy, or cycling).
The authors mention about seed dispersers but the study did not collect any data that supports this idea. It will be necessary to do different approximation in order to explain more accurately the monkey-resource interaction.
Besides, there is not clear what was the total number of focal hours of observations for season, and for age-class. This limits the comparison with some other studies, and with the representativeness of the article.

·

Basic reporting

A generally well presented and clear manuscript. Please see the attached document for details of minor edits.

Experimental design

Generally clear and well defined. Please see the attached document for details that might help to clarify some points.

Validity of the findings

Generally valid although there are two major points that should be addressed to strengthen/correct two of the arguments proposed.These are enclosed within * * charcters on the attached document.

---

## Round 0.2 · Minor Revisions

Please see my comments (minor) attached to the PDF. If you want the Word version, please email me outside the PeerJ system (Hannah.Buckey@lincoln.ac.nz). PeerJ does not allow me to attach a Word doc.

---

## Round 0.3 · accepted · Accept

Please change "determined" on Line 400 to "determine".